# Effects of intra-articular Platelet-Rich Plasma (PRP) injections on osteoarthritis in the thumb basal joint and scaphoidtrapeziotrapezoidal joint

**Elin Swärd**[1,2], **Maria Wilcke**[1,2]*

**1** Department of Clinical Science and Education, Karolinska Institutet, Södersjukhuset, Stockholm, Sweden,
**2** Department for Hand Surgery, Södersjukhuset, Stockholm, Sweden

* maria.wilcke@sll.se

## Abstract

Intra-articular injection of platelet rich plasma (PRP) has been reported to decrease pain and improve function in knee osteoarthritis. There are few reports on the effect of PRP in the treatment of osteoarthritis in the hand. Our aim was to evaluate the effect of PRP-injections on pain and functional outcome in the short-term for osteoarthritis in the thumb basal joint and scaphoidtrapeziotrapezoidal (STT) joint. A retrospective analysis was performed of 29 patients treated with intra-articular PRP injection for painful osteoarthritis in the thumb basal joint (21 patients) or STT joint (eight patients). The patients received two consecutive, radiologically guided PRP injections at an interval of 3–4 weeks. Pain at rest and on load (numerical rating scale (NRS) 0–10), Patient-rated Wrist and Hand Evaluation (PRWHE) score (0–100), grip strength (Jamar) and key pinch were recorded pre-injection and 3 months after the second injection. Mean age was 63 (range 34–86) years and 17 patients were women. We used generalized estimating equations (GEE) to analyze the effect on the outcome variables. Possible predictors were included in the model (high pain level pre-injection, gender, age, manually demanding work, affected joint (thumb base or STT) and use of analgesic). The GEE analysis showed that PRP injections had no effect on reported pain, PRWHE score, grip strength or key pinch. 16/28 patients experience a positive effect according to a yes/no question. The short-term effect of PRP for osteoarthritis in the thumb base and STT-joint is doubtful and needs to be properly investigated in placebo-controlled studies.

## Introduction

Osteoarthritis in the thumb basal joint and scaphoidtrapeziotrapezoidal joint is very common, increases with age and may cause substantial disability in the ageing population [1–3]. Non-operative treatment includes patient education in joint protection, splints, analgesics, and intra-articular corticosteroid injections. Successful conservative treatment that delays or reduces the need for surgery is of great value for both the individual and the public healthcare.

**Data Availability Statement:** All relevant data are within the paper and its Supporting Information files.

**Funding:** The authors received no specific funding for this work.

**Competing interests:** The authors have declared that no competing interests exist.

Intra-articular injection of platelet rich plasma (PRP) has been shown to relieve pain and improve functional outcome compared to hyaluronic acid or saline in knee osteoarthritis [4–7]. PRP is derived by centrifugation of autologous blood to separate the plasma that contains high concentrations of platelets. Platelets contain large number of growth factors and some of these regulate selected biological processes in tissue repair and may have an anti-inflammatory effect. It has been suggested that fibrinogen in PRP may be activated to form a fibrin matrix that may fill cartilage lesions and that PRP may have positive effects on cartilage repair [8].

There are few previous studies on the effect of PRP for osteoarthritis in the hand. One non-comparative pilot study of 10 patients with thumb base osteoarthritis reported decreased pain [9]. A randomized comparison between PRP and intra-articular corticosteroids [10] in a total of 33 patients with thumb base osteoarthritis, concluded that PRP achieved reduced pain and improved function up to 12 months.

We hypothesized that PRP injections would decrease patient-reported pain in the short term for osteoarthritis in the thumb basal joint and STT-joint.

## Materials and methods

This is a retrospective case series based on medical records of all patients that received PRP treatment for radiologically and clinically verified osteoarthritis in the thumb basal joint or STT-joint at a specialized Hand Surgery unit between September 1st 2018 and September 30th 2019 (selection criteria). The Arthrex ACP double-syringe system was used for preparation and injection of the PRP. Fifteen milliliters of venous blood were drawn from each patient and centrifugated at 1500 rpm for five minutes as described by the Arthrex double-syringe manual. The patients received two consecutive PRP injections of 0.5–2 ml at an interval of 3–4 weeks according to recommendations from the manufacturer (Arthrex). The injections were performed at outpatient visits, under sterile conditions and under fluoroscopic guidance, to assure that the PRP was deposited intraarticularly. The patients could resume work and daily activities immediately after the injection as tolerated. There was no specific post-injection hand exercise protocol.

The assessment before and after the PRP injections were standardized. Pain at rest and on load rated by a numerical rating scale (NRS) 0–10 points, Patient-Rated Wrist and Hand Evaluation (PRWHE) score (0–100 points) [11,12], grip strength (Jamar) and key pinch was recorded pre-injection, before the second injection and three months after the second injection (i.e., four months after the first injection). Key pinch and grip strength were measured three times, of which the mean values were recorded. Patients that did not attend the assessment after four months were excluded due to lack of data. There were no other exclusion criteria.

The severeness of the osteoarthritis was classified according to Eaton and Littler [13] for thumb basal osteoarthritis and according to the classification system described by White et al. [14] for STT osteoarthritis.

The study was approved by the Swedish Ethical Review Authority (D.nr 2020–02040).

### Statistics

Pain NRS on load was the primary outcome. Pain NRS scores and PRWE are reported as median (Interquartile range (IQR)) and continuous variables (i.e., key pinch and grip strength) as mean (standard deviation (SD)). In patients with chronic musculoskeletal pain, a reduction of two points in the NRS is considered as a substantial improvement ("much better") by the patients and is recommended as a clinically important outcome [15]. To detect a mean improvement of two points in pain on load NRS (0–10), estimated SD 2 (based on previous

reports [9,16]) (Power 80%, p = 0.05), 10 patients are required. Hence the sample size was regarded as sufficient.

To analyze the repeated measurements, we used generalized estimating equations (GEE) with robust estimator covariances matrix, independent working correlation matrix (according to lowest QICC) and a linear model for all variables. High pre-injection pain level ($\geq$5 pain at rest, based on the 75th quartile), gender, age, manually demanding work, affected joint (thumb base or STT), and use of analgesic were included as possible predictors in the model. The changes in outcome variables over time and the effects on the predictors are presented as beta coefficient ($\beta$) with 95%CI. The statistical software used was SPSS® version 28.

## Results

Thirty-three patients had received PRP treatment. Four patients did not attend the 3-months visit and were excluded. The mean age of the remaining 29 patients was 63 (range 34–86) years. Seventeen patients were women. Twelve patients were retired, one patient was on sick leave (for unrelated reasons). Of the 16 work-active patients, four patients had administrative work and 12 manually demanding work. Twenty-one patients had thumb basal joint osteoarthritis and eight patients had STT-joint osteoarthritis. Table 1 presents the grading of the osteoarthritis.

Twenty-two patients had used a splint prior to the PRP treatment, and 20 patients had previously tried corticosteroid injections. Two patients were prescribed analgesics for pain in the thumb base or STT-joint at/in conjunction with the first injection, seven patients frequently used analgesics for unrelated causes and 20 patients were not prescribed any analgesics during the study period.

The values of the outcome variables at the different assessments are presented in Table 2.

Table 3 presents the results of the GEE analyses. A high pre-injection pain level predicted more pain at rest and on load at follow-up ($\beta$ = 4.0(3.0–4.9) and 1.9(0.8–2.9), respectively). When corrected for pre-injection pain level, there was no significant improvement of pain at rest or on load after PRP injection. A high pre-injection pain level and age 65 years or older predicted a worse PRWHE score ($\beta$ = 22(12–31) and ($\beta$ = 12(2–22), respectively) but the PRP injection did not affect PRWHE score. A high pre-injection pain level, age over 65 years and female gender predicted lesser grip strength ($\beta$ = -5(-12- -24), -8(-14- -2) and -18(-12- -24), respectively) and key pinch ($\beta$ = -1.8(-2.7–1.0), -3.3(-4.9- -1.6) and -2.6(-1.1–4.1) respectively). There was a significant decrease of grip strength at the second PRP injection but not at the final assessment and there was no change in key pinch after PRP injection. Manually demanding work, affected joint, and use of analgesic did not affect the outcome variables.

Of 28 patients that had answered the question if they experienced a positive effect of the injections, 16 answered "yes", 9 "no" and 3 answered "I don't know". 15 patients requested a

**Table 1. Localization and grading of the osteoarthritis.**

| Localization of osteoarthritis | Grade | n = |
|---|---|---|
| Thumb basal joint | Eaton 1 | 2 |
| | Eaton 2 | 4 |
| | Eaton 3 | 12 |
| | Eaton 4[1] | 3 |
| Scaphoidtrapeziotrapezoidal joint | White 1 | 1 |
| | White 3 | 7 |

[1]No symptoms from the STT-joint.

**Table 2. Pain, patient-rated results and strength.**

|  | Pain at rest NRS (0–10) | Pain on load NRS (0–10) | PRWHE (0–100) | Grip strength (kg) | Key pinch (kg) |
|---|---|---|---|---|---|
| Preinjection | 2 (1–5) | 8 (6–9) | 65 (55–77) | 23 (13) | 5.5 (2.5) |
| 2nd injection | 0 (0–4) | 7 (4–10) | 60 (43–77) | 22 (13) | 5.8 (2.9) |
| 3 months after 2nd injection | 1 (0–3) | 6 (4–9) | 54 (40–81) | 23 (14) | 6.0 (2.9) |

Pain and PRWHE values are presented as median (IQR). Grip strength and key pinch values are presented as mean (SD).

new PRP injection, and two patients wished for a corticosteroid injection at the follow-up 4 months after the first injection. There were no recorded infections or other adverse events.

## Discussion

We could not find a significant effect on patient-reported pain, hand disability or strength in the short-term after intra-articular PRP injections for osteoarthritis in the thumb basal joint or STT-joint. On a yes/no question of experienced improvement, more than half of the patients said yes. However, even when possible confounders were not taken into account, the mean individual reduction in pain NRS did not amount to two points (Table 2), corresponding to a substantial improvement [15].

There are few reports of the effect of PRP on osteoarthritis in the hand. Malahias el al. [10] performed a blinded randomized comparison between two PRP injections with a two week's interval (16 patients) and intra-articular corticosteroid injections (17 patients) for thumb base osteoarthritis and reported a decrease in pain visual analogue scale (VAS) from 75 to 40 (out of 100) and improved quick Disabilities of the Arm, Shoulder and Hand (qDASH) score from 50 to 33 (out of 100) at 3 months in the PRP group. In comparison, the corticosteroid group also reported better VAS (from 70 to 20 out of 100) at 3 months but after 12 months, the PRP group reported significantly less pain and better function than the corticosteroid group. The reported improvement in pain and quickDASH almost are almost in line with trapeziectomy after one year [16], hence a result very dissimilar to ours. Malahias et al. used PRP that was prepared manually at a local hospital laboratory, while we used a commercial kit (Arthrex ACP double-syringe system). Therefore, the concentration of platelets may differ.

Loibl et al. [9] evaluated 10 patients with thumb basal joint osteoarthritis treated with two intra-articular PRP (Arthrex ACP double-syringe system) injections 4 weeks apart and reported decreased pain with activity (VAS) from 6.2 to 4.0 after three months and improvement in the Disabilities of the arm, shoulder, and hand (DASH) score from 33 to 20 (out of 100) but unaffected grip strength and pinch strength. After 6 months, VAS had increased to 5.5 and DASH to 27.

**Table 3. Effect of PRP injection on pain, patient-rated result and strength.**

|  | Beta coefficients for improvement (95%CI) | | | | |
|---|---|---|---|---|---|
|  | Pain at rest NRS* | Pain on load NRS* | PRWHE ** | Grip strength*** | Key pinch*** |
| Preinjektion - 2nd injection | 0.61(-0.43–1.66) | -0.73(-1.8–0.37) | 2(-10-10) | -2.86(-5.44–0.02) | -0.24(-0.94–0.47) |
| Preinjektion—3 months after 2nd injection | 0.44(-0.50–1.38) | -0.98(-2.17–0.20) | 0(-5-10) | -1.72(-5.08–1.64) | 0.10(-0.85–0.97) |

*corrected for high pain level pre-injection.

**corrected for high pain level pre-injection and age.

*** corrected for high pain level pre-injection, age and gender.

Beta coefficient: Expected population average change of the outcome variable between the assessments.

One possible explanation to the fact that our result opposes Malahias et al. and Lobli et al. could be that the latter studies used rather simple statistical analyses and have not corrected the results for confounding factors such as pre-injection pain level, gender, age etc. Malahias et al. has a randomized design which may compensate for potential confounders. However, PRP is not compared to placebo but to corticosteroid and one should be aware of that, despite widely used, there is lack of evidence that corticosteroid injections have effect on pain and function compared to placebo for thumb basal joint osteoarthritis [17] as well as in other joints [18]. Hence, the observed effect in the previous studies could be placebo. It has been shown that placebo is effective in the treatment of osteoarthritis, especially for pain, stiffness, and self-reported function and the placebo for new treatments (like PRP) tend to be more effective [19]. The placebo effect increases with increase baseline pain severity [19]. In this study, we found that the pain level before PRP treatment had the greatest impact on the outcome variables. Hence it is a strength of the study that pain level before injection is corrected for in the analysis. In addition to a potential placebo effect, chance regression to the mean and natural disease remission are possible explanations to the previous observed results of PRP injections that could not be reproduced in our study.

One can question why our study showed no effect of PRP for thumb basal and STT-joint osteoarthritis when placebo-controlled studies have shown that PRP has effect for knee osteoarthritis [4,5]. We have no plausible explanation, and we suggest that this study alone should not be used to dismiss PRP injections for osteoarthritis in the hand, but the effect needs to be further and properly investigated in placebo-controlled studies.

Psychological factors may have an impact on pain outcome. Anxiety, depression, and pain catastrophizing has been shown to predict poor outcome after hand surgery [20,21]. It is reasonable that psychological factors similarly may affect the result of injection treatment. A high rated pain at rest in patients with osteoarthritis can some extent be considered a proxy for pain catastrophizing and is therefore to some degree included as predictor in the analysis. However, a weakness of the study is that we have not been able to correct the results for depressions and anxiety.

This study has several other limitations. Obviously, it is a retrospective observational study with associated risk for uncontrolled biases. The data was collected from medical records and lack forinformation about some factors that may affect PROMs. For example, we have no information on the extent to which patients used splints and of over-tha-counter analgesics post-injection. Further, the studied sample is relatively small. However, it is not small compared to the previous studies discussed [9,10] that, has shown a significant effect of PRP, hence it is not likely that the sample size explains the absence of a detectable effect in this study. Another limitation is the short follow-up. Hypothetically there could a late-onsetting effect. However, we consider that scenario unlikely.

Unlike intra-articular corticosteroid injections that may have a harmful effect on cartilage [22,23], PRP have no documented side effects [6,8] and does not increase the risk of adverse events compared with injection of hyaluronic acid or saline [24]. PRP is not classified as a drug and its use is therefore not restricted by any regulations. Consequently, there has been a flurry of various applications of this harmless and potentially salutary injection, from musculoskeletal disorders to cosmetics. Hence, the risk with PRP is rather an overuse of a costly and perhaps ineffective treatment and therefore it is important to establish if there is a true therapeutic effect. The result of this study contradicts the hypothesis that PRP injections have an effect in terms of pain relief, disability, or strength for osteoarthritis in the thumb basal joint or STT-joint.

## Supporting information

**S1 Dataset.**
(XLSX)

## Author Contributions

**Conceptualization:** Maria Wilcke.

**Data curation:** Maria Wilcke.

**Formal analysis:** Elin Swärd, Maria Wilcke.

**Investigation:** Maria Wilcke.

**Methodology:** Maria Wilcke.

**Project administration:** Maria Wilcke.

**Writing – original draft:** Elin Swärd, Maria Wilcke.

**Writing – review & editing:** Elin Swärd, Maria Wilcke.

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
