## [Decision Letter · Decision Letter 0]

15 Sep 2021

PONE-D-21-21631Effects of intra-articular Platelet-Rich Plasma (PRP) injections on osteoarthritis in the thumb basal joint and scaphoidtrapeziotrapezoidal joint.PLOS ONE

Dear Dr. Wilcke,

Thank you for submitting your manuscript to PLOS ONE. After careful consideration, we feel that it has merit but does not fully meet PLOS ONE’s publication criteria as it currently stands. Therefore, we invite you to submit a revised version of the manuscript that addresses the points raised during the review process.

We look forward to receiving your revised manuscript.

Kind regards,

Gabriel de Araújo, M.D., MSc

Academic Editor

PLOS ONE

Journal Requirements:

2. Please ensure that you include a title page within your main document. You should list all authors and all affiliations as per our author instructions and clearly indicate the corresponding author

Reviewers' comments:

Reviewer's Responses to Questions

**Comments to the Author**

1. Is the manuscript technically sound, and do the data support the conclusions?

Reviewer #1: Partly

Reviewer #2: Partly

2. Has the statistical analysis been performed appropriately and rigorously? 

Reviewer #1: No

Reviewer #2: I Don't Know

3. Have the authors made all data underlying the findings in their manuscript fully available?

Reviewer #1: Yes

Reviewer #2: Yes

4. Is the manuscript presented in an intelligible fashion and written in standard English?

Reviewer #1: Yes

Reviewer #2: Yes

5. Review Comments to the Author

Reviewer #1: This is a relatively short and simple analysis of available data to answer whether PRP injections improve osteoarthritis in the thumb basal joint and scaphoidtrapeziotrapezoidal joint. The organization is clear and the short paper is well written. I submit the findings may be potentially useful but it is a one-arm study which is also much more open to bias and misleading conclusions. A main question I have is whether there is enough substantial material for a publication given the paper is relatively much shorter than others I have refereed in the journal and the data is very simply analyzed.

I focus on the statistical aspects for the study. Overall, I find the analysis relatively simple which can be particularly problematic when the sample size is small. Some particular concerns are

a) There was mention about excluding patients from the study because they had incomplete data. It is not clear what exclusion rule was used whether the missingness had something to do with the disease progression or the pain the patient is suffering. Are data missing at random or completely missing at random or otherwise? Statistical methods will then have to specially used for different types of missingness;

b) There are statistical tests that are more reliable for analysis small sample data that are more reliable; permutation tests may be better alternatives. Similarly, Shapiro-Wilk’s test results on whether the data are normally distributed can also be questionable for small data set;

and

c) There were demographic data or covariates on the patients but there was no discussion if some of them play a role in explaining the findings the authors obtained.

Reviewer #2: Well written and pleasant to read paper.

Introduction:

- How is it relevant to only focus on the short-term effects of this treatment? I'm not sure it is, but please explain. As CMC1 OA is a chronic condition (we will all get osteoarthritis of the thumb if we live long enough), we should focus on treatments that have long-term benefits. Most likely a "quick fix" with an injection or surgical intervention is not the solution.

- Please change "conservative treatment" to "non-operative treatment".

Methods:

- The 3-4 weeks interval between the injections: please explain how you chose this time interval.

- Symptom intensity of CMC1 OA is strongly related to the influence of psychosocial aspects; patients with greater symptoms of catastrophic thinking and depression have greater pain intensity. In many multivariable analysis, these are the strongest contributors to several outcomes including pain (see papers of Becker et al., Wilkens et al., or Prof. Ring). At least, this must be mentioned as it could lead to a potential bias in the findings of the current paper.

- How did you estimate an SD of 2 points for NRS pain?

- What where the inclusion criteria or selection criteria for patients to receive a PRP injection?

Results & Discussion:

- Please write out "thirty-three" and other numbers that are at the beginning of a sentence.

- Line 111-112: these results don't exceed the MCID. Although there is some variety among studies about the MCID for pain scores (NRS/VAS), most agree that an improvement of 1 point is not enough to reach an MCID in outcome (Danoff 2018, Lee 2003). Even the study referred to in this manuscript (Salaffi et al.) supports the use of a "much better" improvement on pain relief as a clinically important outcome: and that is 2 points. I think the authors must revise this point and their conclusion. They found a significant but not a clinically important difference in pain improvement. This is in line with the other study findings that shows no difference in PROMs. And it does not support the (costly) use of PRP for patients with CMC1 OA seeking specialty care.

6. PLOS authors have the option to publish the peer review history of their article (what does this mean?). If published, this will include your full peer review and any attached files.

Reviewer #1: No

Reviewer #2: No

---

## [Author Response · Author response to Decision Letter 0]

6 Oct 2021

Thank you for a valuable review. Below we address the reviewers’ concerns point by point.

Reviewer #1: 

I focus on the statistical aspects for the study. Overall, I find the analysis relatively simple which can be particularly problematic when the sample size is small. 

We have addressed the reviewer’s concern about too simple statistical analysis and with help from a biomedical statistician, we have remade the analysis with a GEE model. This made a significant difference since the formerly significant improvement of pain was no longer significant after the confounding predictors were considered in the analysis. Hence, the conclusion and a major part of the discussion is revised.

 Some particular concerns are

a) There was mention about excluding patients from the study because they had incomplete data. It is not clear what exclusion rule was used whether the missingness had something to do with the disease progression or the pain the patient is suffering. Are data missing at random or completely missing at random or otherwise? Statistical methods will then have to specially used for different types of missingness; 

Four patients had to be excluded since they did not attend the 3 months assessment. We now write this more clearly in line 62-64. The former wording was unclear. The reason for not attending could not be found in the medical records. We consider them missing at random. After ha have redone the analyses with correction for predictors (please see below), a potential difference in the excluded patient compared to the analysed sample, should not be a problem.

b) There are statistical tests that are more reliable for analysis small sample data that are more reliable; permutation tests may be better alternatives. Similarly, Shapiro-Wilk’s test results on whether the data are normally distributed can also be questionable for small data set; 

We now use GEE for analyses instead so this should not be an issue.

and

c) There were demographic data or covariates on the patients but there was no discussion if some of them play a role in explaining the findings the authors obtained. 

The covariates/possible predictors are now included in a GEE model.

Reviewer #2: 

Well written and pleasant to read paper.

Introduction:

- How is it relevant to only focus on the short-term effects of this treatment? I'm not sure it is, but please explain. As CMC1 OA is a chronic condition (we will all get osteoarthritis of the thumb if we live long enough), we should focus on treatments that have long-term benefits. Most likely a "quick fix" with an injection or surgical intervention is not the solution. 

We believe that the short-term aspect is relevant since injections are assumed to foremost have a short-term effect. We did not have long-term data and we agree that the short-term follow-up is a limitation and mention this in the discussion line 173-174. 

- Please change "conservative treatment" to "non-operative treatment". 

This is now changed.

Methods:

- The 3-4 weeks interval between the injections: please explain how you chose this time interval. 

There is no consensus or scientific base for the number of PRP injections or the optimal interval. The patients had injections with 3-4 weeks based on recommendations from Arthrex. Due to the retrospective design this was not part of the study design.

- Symptom intensity of CMC1 OA is strongly related to the influence of psychosocial aspects; patients with greater symptoms of catastrophic thinking and depression have greater pain intensity. In many multivariable analysis, these are the strongest contributors to several outcomes including pain (see papers of Becker et al., Wilkens et al., or Prof. Ring). At least, this must be mentioned as it could lead to a potential bias in the findings of the current paper. 

We agree with the reviewer and we have no included high pre-injection pain as a predictor in the analysis. The issue is also discussed in line 162-166. Although the above mentioned authors have produced many interesting articles, we could not find a suitable reference from these specific authors for this particular issue.

- How did you estimate an SD of 2 points for NRS pain? 

 SD 2 was estimated based on patient-rated pain in a large sample of patients planned for surgery for thumbbase osteoarthritis (Wilcke et al.) with a mean pain on load of 76 (SD 17) of 100 and the small study of Loibl et al; pain VAS 4.0 (SD 2.4) at 3 months after a PRP injection. This is now stated more clearly in line 76-77. In retrospect, the estimation was a bit low since the SD (which we do not present in the manuscript that reports median and IQR) of pain on load after 3 months were 3 corresponding to a required sample of 20 so we don’t think the study is underpowered.

- What where the inclusion criteria or selection criteria for patients to receive a PRP injection? 

The selection criteria were osteoarthritis in the thumbbase or STT-joint. This is now pointed out in line 51.

Results & Discussion:

- Please write out "thirty-three" and other numbers that are at the beginning of a sentence. 

This is now corrected.

- Line 111-112: these results don't exceed the MCID. Although there is some variety among studies about the MCID for pain scores (NRS/VAS), most agree that an improvement of 1 point is not enough to reach an MCID in outcome (Danoff 2018, Lee 2003). Even the study referred to in this manuscript (Salaffi et al.) supports the use of a "much better" improvement on pain relief as a clinically important outcome: and that is 2 points. I think the authors must revise this point and their conclusion. 

According to reviewer #1:s concern about analyses being too simple, we have now remade them I cooperation with a statistician using a GEE model for longitudinal data with the possibility to correct for possible predictors. Interesting, this analysis showed no significant impact of PRP injection. Hence the conclusion and discussion are quite extensively revised.

---

## [Decision Letter · Decision Letter 1]

6 Dec 2021

PONE-D-21-21631R1Effects of intra-articular Platelet-Rich Plasma (PRP) injections on osteoarthritis in the thumb basal joint and scaphoidtrapeziotrapezoidal joint.PLOS ONE

Dear Dr. Wilcke,

Thank you for submitting your manuscript to PLOS ONE. After careful consideration, we feel that it has merit but does not fully meet PLOS ONE’s publication criteria as it currently stands. Therefore, we invite you to submit a revised version of the manuscript that addresses the points raised during the review process.

We look forward to receiving your revised manuscript.

Kind regards,

Gabriel de Araújo, M.D., MSc

Academic Editor

PLOS ONE

Reviewers' comments:

Reviewer's Responses to Questions

**Comments to the Author**

1. If the authors have adequately addressed your comments raised in a previous round of review and you feel that this manuscript is now acceptable for publication, you may indicate that here to bypass the “Comments to the Author” section, enter your conflict of interest statement in the “Confidential to Editor” section, and submit your "Accept" recommendation.

Reviewer #2: All comments have been addressed

Reviewer #3: (No Response)

2. Is the manuscript technically sound, and do the data support the conclusions?

Reviewer #2: Partly

Reviewer #3: No

3. Has the statistical analysis been performed appropriately and rigorously? 

Reviewer #2: Yes

Reviewer #3: No

4. Have the authors made all data underlying the findings in their manuscript fully available?

Reviewer #2: Yes

Reviewer #3: Yes

5. Is the manuscript presented in an intelligible fashion and written in standard English?

Reviewer #2: Yes

Reviewer #3: Yes

6. Review Comments to the Author

Reviewer #2: I would recommend to accept this paper. However, one more suggestion:

* The conclusion of this paper is that PRP injections do not lead to pain relief, decrease of physical limitations or improvement of strength. The last paragraph of the discussion however, seems like a plea for the harmless use of PRP injections. Please look at this again. And delete the last sentence: "Although this study has methodological flaws, and it is important to test PRP in randomized placebo-controlled studies, our study results suggest that PRP injections have no effect on pain intensity, physical limitations, or thumb strength among patients with osteoarthritis in the thumb basal joint or STT-joint."

Reviewer #3: Thank you for the opportunity to review this paper.

The present paper is a 29-patient retrospective case series on the use of PRP in the treatment of thumb basal joint and STT joint OA. The authors observed a significant but small effect in terms of pain relief for osteoarthritis in the thumb basal joint or STT-joint.

The paper is in general well written and easy to follow. However, it has several methodological issues.

I credit the authors for prospective data collection. However, I cannot see why the authors did not define clear study objective/question and a study protocol a priori in order to collect a prospective case series. This leads to several shortcomings.

Perhaps the biggest concern is the lack of explicit inclusion and/or exclusion criteria which leads to major heterogeneity between participants: Both thumb basal joint and STT arthrosis (in my opinion, these conditions may be related to different biomechanical changes), age 34-86, posttraumatic or primary arthrosis (?), the duration of symptoms before treatment, pre-treatment pain intensity (no borders for inclusion/exclusion), associated conditions that might affect the result (e.g. small joint arthritis in general, thumb MCP arthritis), previous trauma or surgery of the hand/wrist and treatment of contralateral TMC/STT joint arthritis (successful non-operative/operative treatment of contralateral TMC/STT might affect patients expectations).

While the authors have assessed the severity of the arthritic changes - in my opinion it is as important to assess the joint congruency as the thumb basal joint might be clinically stable, instable or chronically subluxated - which might affect the outcome of the injection as the biomechanical basis of the disease might be different.

Another concern lies in the preparation of PRP - how rich is platelet-rich? No laboratory testing is made to control for the quality of the agent. Moreover, how do the authors believe that the quantity (0,5-2ml) of PRP affects the outcome? How was the interval between treatments decided? The regulatory aspects of the use of PRP for TMC and STT OA should also be declared.

The enrollment lasted from sep 2018 to sep 2019. Was it a consecutive series? Was the enrollment based on patient preference? How many patients declined? Were there any exclusion criteria?

Post-treatment regimen? The authors state that “The patients could resume work and daily activities immediately after the injection as tolerated.” - did the patients use a splint? Was there any hand exercise regimen? The authors report that two patients were prescribed pain medication at first injection, whereas 20 did not use any pain medication. Seven patients used pain medication for “unrelated reasons” All of these might significantly affect the outcome as the post-treatment protocol was not defined.

In the statistics paragraph the authors have presented a power calculation. A post-hoc power calculation seems off, why not just look at 95% CI? Why not repeated measures analysis of variance?

The results are very brief and mainly presented without numbers (percentage, SD, IQR, 95% CI) or significance levels. Lines 111-114 are not results per se but patient characteristics (a methodological concern). The minimally clinically important difference for pain presented in power calculation is 2 points (2 SD). However, the authors interpret a 1-point decrease in NRS a “substantial improvement”. How is this clinically relevant? The patients reported the pain to be “slightly better” - which could represent treatment bias and reflect the expectations of the patients. Patient acceptable symptom state (PASS) would better reflect the clinical outcome as PASS is the highest symptom level at which patients consider themselves well.

Moreover, the authors report that 15 patients requested a new PRP injection, and two patients wished for a corticosteroid injection at the follow-up 4 months after the first injection. Thus, at least 58% of the patients were probably not happy with the treatment. Was surgery an option? What about the 42%, what happened then? I would be interested if the treatment reduces the number of surgeries in the long-term. As the authors point out in the introduction - successful conservative treatment that delays or reduces the need for surgery is of great value. However, the present paper does not give any clinically relevant information on the matter as the results are very short term.

The subject of the present manuscript is very welcome as an increasing interest in biologic agents - as nonoperative treatment modalities or to augment surgical procedures - has been observed in recent years. However, the methodological shortcomings of the present paper make it hard to support the publication of this paper.

7. PLOS authors have the option to publish the peer review history of their article (what does this mean?). If published, this will include your full peer review and any attached files.

Reviewer #2: No

Reviewer #3: No

---

## [Author Response · Author response to Decision Letter 1]

10 Dec 2021

Reviewer #2: I would recommend to accept this paper. However, one more suggestion:

* The conclusion of this paper is that PRP injections do not lead to pain relief, decrease of physical limitations or improvement of strength. The last paragraph of the discussion however, seems like a plea for the harmless use of PRP injections. Please look at this again. And delete the last sentence: "Although this study has methodological flaws, and it is important to test PRP in randomized placebo-controlled studies”, our study results suggest that PRP injections have no effect on pain intensity, physical limitations, or thumb strength among patients with osteoarthritis in the thumb basal joint or STT-joint."

According to the reviewer’s suggestion, we have deleted the last sentence “Although this study has methodological flaws, and it is important to test PRP in randomized placebo-controlled studies.” 

Reviewer #3: Thank you for the opportunity to review this paper.

I credit the authors for prospective data collection. However, I cannot see why the authors did not define clear study objective/question and a study protocol a priori in order to collect a prospective case series. This leads to several shortcomings.

We agree with the reviewer that a prospective case-series would have been preferable. However, this is a retrospective study based on medical protocols.

Perhaps the biggest concern is the lack of explicit inclusion and/or exclusion criteria which leads to major heterogeneity between participants: Both thumb basal joint and STT arthrosis (in my opinion, these conditions may be related to different biomechanical changes), age 34-86, posttraumatic or primary arthrosis (?), the duration of symptoms before treatment, pre-treatment pain intensity (no borders for inclusion/exclusion), associated conditions that might affect the result (e.g. small joint arthritis in general, thumb MCP arthritis), previous trauma or surgery of the hand/wrist and treatment of contralateral TMC/STT joint arthritis (successful non-operative/operative treatment of contralateral TMC/STT might affect patients expectations).

We agree that this is a weakness with the retrospective design as discussed in line 169-176. On the other hand, the wide inclusion criteria make the result generalizable and there is no comparison between groups that would be biased by the heterogeneity.

While the authors have assessed the severity of the arthritic changes - in my opinion it is as important to assess the joint congruency as the thumb basal joint might be clinically stable, instable or chronically subluxated - which might affect the outcome of the injection as the biomechanical basis of the disease might be different.

This is an interesting point. However, since joint stability in thumb base osteoarthritis was not a variable recorded in the medical records, the effect if the joint stability could not be assessed.

Another concern lies in the preparation of PRP - how rich is platelet-rich? No laboratory testing is made to control for the quality of the agent. 

Testing the concentration is not possible given the retrospective design. This is a pragmatic study of the effect of PRP prepared with a commercial kit. As the reviewer hints, the concentration may theoretically affect the effect of PRP, and this is discussed in line 132-34.

Moreover, how do the authors believe that the quantity (0,5-2ml) of PRP affects the outcome? 

We have not included injected quantity as a covariate in the analysis since two different joints with different sizes (CMC1 and STT) were treated and we believe that quantity would be difficult to assess as a covariate. Therefore, we cannot say anything about the potential effect of the injected quantity. 

How was the interval between treatments decided? The regulatory aspects of the use of PRP for TMC and STT OA should also be declared.

The interval of interval of 3-4 weeks was recommended by the manufacturer (Artrex). This is now stated in line 52. There are no regulatory aspects of PRP as discussed in line 180-1.

The enrollment lasted from sep 2018 to sep 2019. Was it a consecutive series? Was the enrollment based on patient preference? How many patients declined? 

As described in line 45-48, all patients that received PRP treatment for radiologically and clinically verified osteoarthritis in the thumb basal joint or STT-joint between September 1st2018 and September 30th2019 were included. This is not a prospective study and hence, there was no invitation to participate in a study. 

Were there any exclusion criteria?

Exclusion criteria was lack of data as describes in line 62-63 “Patients that did not attend the assessment after four months were excluded due to lack of data.“. We have added the sentence “There were no other exclusion criteria.” To clarify this.

Post-treatment regimen? The authors state that “The patients could resume work and daily activities immediately after the injection as tolerated.” - did the patients use a splint? 

We have no information of whether and to what extent the patients used splints. This is now mentioned as a weakness in the discussion line 169-73.

Was there any hand exercise regimen? The authors report that two patients were prescribed pain medication at first injection, whereas 20 did not use any pain medication. Seven patients used pain medication for “unrelated reasons” All of these might significantly affect the outcome as the post-treatment protocol was not defined.

There was no post-injection hand exercise protocol. This is now stated in line 55-56. The use of pain killers was included as a possible predictor in the model. We have added the sentence “Manually demanding work, affected joint, and use of analgesic did not affect the outcome variables.” in line 109-110 to clarify that these included predictors in the model did not have an effect.

In the statistics paragraph the authors have presented a power calculation. A post-hoc power calculation seems off, why not just look at 95% CI? Why not repeated measures analysis of variance? The power calculation was performed prior to the analysis to estimate if there were enough available patients to enable a meaningful evaluation of this sample. 

GEE is more efficient and flexible than Repeated measures ANOVA. GEE is an extension of generalized linear models such as RM-ANOVA to cover repeated measures and other forms of dependent data and can handle categorical outcome; non-normal and non-linear data.

The results are very brief and mainly presented without numbers (percentage, SD, IQR, 95% CI) or significance levels. 

Could it be that the reviewer has read the first version and not the revised manuscript? Line 99-109 are all results with numbers; beta coefficient (β) with 95%CI.

Lines 111-114 are not results per se but patient characteristics (a methodological concern). 

Is this the correct lines or do they refer to the non-revised manuscript? Lines 111-114: 

“months after the first injection. There were no recorded infections or other adverse events.

Discussion

We could not find a significant effect on patient-reported pain, hand disability or strength in”

In the original, non-revised manuscript line 111-114 was: 

“22 patients had used a splint prior to the PRP treatment, and 20 patients had previously tried corticosteroid injections. Two patients were prescribed analgesics for pain in the thumb base or STT-joint at/in conjunction with the first injection, 7 patients frequently used analgesics for unrelated causes and 20 patients were not prescribed any analgesics during the study period.”

If the reviewer addresses the lines in the original manuscript, we do not really agree. This is a description of the sample and that type of information is usually presented in result and we do not think that this information belongs under Methods. 

The minimally clinically important difference for pain presented in power calculation is 2 points (2 SD). However, the authors interpret a 1-point decrease in NRS a “substantial improvement”. How is this clinically relevant? 

We do not recognize this from the current manuscript, and we are concerned that the reviewer refers to the original and not the current revised manuscript?

Line 72-74: In patients with chronic musculoskeletal pain, a reduction of two points in the NRS is considered as a substantial improvement (“much better”) by the patients and is recommended as a clinically important outcome [15].

Line 119-121: However, even when possible confounders were not taken into account, the mean individual reduction in pain NRS did not amount to two points (table 2), corresponding to a substantial improvement [15].

The patients reported the pain to be “slightly better” - which could represent treatment bias and reflect the expectations of the patients. Patient acceptable symptom state (PASS) would better reflect the clinical outcome as PASS is the highest symptom level at which patients consider themselves well.

Again, we do not recognize this from the current manuscript, and we are concerned that the reviewer refers to the original and not the revised manuscript?

PASS seems like a good and relevant concept and a good a complementary measure to MCID but this was not used in the medical records and as to our knowledge there are no published cut-off value for osteoarthritis in the hand/wrist.

Moreover, the authors report that 15 patients requested a new PRP injection, and two patients wished for a corticosteroid injection at the follow-up 4 months after the first injection. Thus, at least 58% of the patients were probably not happy with the treatment. Was surgery an option? What about the 42%, what happened then? I would be interested if the treatment reduces the number of surgeries in the long-term. As the authors point out in the introduction - successful conservative treatment that delays or reduces the need for surgery is of great value. However, the present paper does not give any clinically relevant information on the matter as the results are very short term.

We cannot provide any information about further treatment including operation. This study does not give any information about the long-term effect in terms om reduced number of operation but only the short-term results as stated in line 5, 41, 117, 176.

The subject of the present manuscript is very welcome as an increasing interest in biologic agents - as nonoperative treatment modalities or to augment surgical procedures - has been observed in recent years. However, the methodological shortcomings of the present paper make it hard to support the publication of this paper.

We agree that this study has shortcomings. But at this time when PRP is widely administered despite limited evidence of its effect and reports may be affected by commercial interests, we believe that it is important to publish all available clinical results.

---

## [Decision Letter · Decision Letter 2]

7 Feb 2022

Effects of intra-articular Platelet-Rich Plasma (PRP) injections on osteoarthritis in the thumb basal joint and scaphoidtrapeziotrapezoidal joint.

PONE-D-21-21631R2

Dear Dr. Wilcke,

We’re pleased to inform you that your manuscript has been judged scientifically suitable for publication and will be formally accepted for publication once it meets all outstanding technical requirements.

Kind regards,

Gabriel de Araújo, M.D., MSc

Academic Editor

PLOS ONE

Additional Editor Comments (optional):

The manuscript is well written and value. Besides that, I agree that all reviewers comments would improve the scientific quality of you study. For this reason, I am sending the reviewers suggestions for your appreciation.

Reviewers' comments:

Reviewer's Responses to Questions

**Comments to the Author**

1. If the authors have adequately addressed your comments raised in a previous round of review and you feel that this manuscript is now acceptable for publication, you may indicate that here to bypass the “Comments to the Author” section, enter your conflict of interest statement in the “Confidential to Editor” section, and submit your "Accept" recommendation.

Reviewer #4: (No Response)

Reviewer #5: (No Response)

2. Is the manuscript technically sound, and do the data support the conclusions?

Reviewer #4: Partly

Reviewer #5: Yes

3. Has the statistical analysis been performed appropriately and rigorously? 

Reviewer #4: Yes

Reviewer #5: Yes

4. Have the authors made all data underlying the findings in their manuscript fully available?

Reviewer #4: Yes

Reviewer #5: Yes

5. Is the manuscript presented in an intelligible fashion and written in standard English?

Reviewer #4: Yes

Reviewer #5: Yes

6. Review Comments to the Author

Reviewer #4: Thank you for the possibility to review this manuscript. As the authors report, there are few studies of the effect of PRP on osteoarthritis in the hand. This could be interesting work. However, some minor issues should be addressed.

The results section could be further explored by the authors.

The data described in lines 87 to 98 could be presented in a table, including the respective percentage of each characteristic.

Line 95 - The timing of corticosteroid use prior to PRP treatment may affect results. Considering that the majority of patients used corticosteroids (n=20), it is necessary to describe the period of time of use before treatment with PRP.

Table 2, considering the small number of the sample and the variability in the scores, it is important to present the percentage of improvement among the patients. Another strategy is to present the average of the differences between the moments.

Line 112- The authors described that 16 patients (57%) answered “yes” to the question whether they experienced a positive effect from the injections. It is necessary to explore who these patients are. Are there characteristics associated with this response (age, sex...)?

The subject of the present study is interesting and “polemical”, since there is little evidence on the subject. The subject of the present study is interesting and controversial, since there is little evidence on the subject. However, considering the limitations of the study (sample number and retrospective data), a detailed description of the results is necessary.

Reviewer #5: The Authors present a retrospective analysis about PRP injection for CMC-I or STT osteoarthritis.

The topic is interesting and relevant for the clinicians. The manuscript is well written and easy to read. The corrected version already take into account the previous comments.

The only two remaining problems that cannot bee changed are:

- the real composition of the injected PRP. It was obtained with a commercial kit and there is no way to check what was injected. In my opinion the technique is well described and could b reproduced easily.

- the amount injected. It is often difficult to inject a significant amount in the CMC-I and STT due to the small joint space and the arthritic changes and it is possible that the efficacy is dose-dependent. In the knee, a joint with a much larger intrarticular volume, the positive effect could be demonstrated. Future studies should address this issue.

Despite these two points I recommend to accept the paper.

7. PLOS authors have the option to publish the peer review history of their article (what does this mean?). If published, this will include your full peer review and any attached files.

Reviewer #4: No

Reviewer #5: No

---

## [Editor Report · Acceptance letter]

28 Feb 2022

PONE-D-21-21631R2 

Effects of intra-articular Platelet-Rich Plasma (PRP) injections on osteoarthritis in the thumb basal joint and scaphoidtrapeziotrapezoidal joint. 

Dear Dr. Wilcke:

I'm pleased to inform you that your manuscript has been deemed suitable for publication in PLOS ONE. Congratulations! Your manuscript is now with our production department. 

Kind regards, 

on behalf of

Professor Gabriel de Araújo 

Academic Editor

PLOS ONE